# Topology Optimization for FDM Parts Considering the Hybrid Deposition Path Pattern

**DOI:** 10.3390/mi11080709

**Published:** 2020-07-22

**Authors:** Shuzhi Xu, Jiaqi Huang, Jikai Liu, Yongsheng Ma

**Affiliations:** 1Department of Mechanical Engineering, University of Alberta, Edmonton, AB T2G 2G8, Canada; shuzhi@ualberta.ca; 2Center for Advanced Jet Engineering Technologies (CaJET), Key Laboratory of High Efficiency and Clean Mechanical Manufacture (Ministry of Education), School of Mechanical Engineering, Shandong University, Jinan 250100, China; jiaqih@mail.sdu.edu.cn (J.H.); jikai_liu@sdu.edu.cn (J.L.)

**Keywords:** solid orthotropic material with penalization, hybrid deposition paths, double smoothing and projection, fused deposition modeling

## Abstract

Based on a solid orthotropic material with penalization (SOMP) and a double smoothing and projection (DSP) approach, this work proposes a methodology to find an optimal structure design which takes the hybrid deposition path (HDP) pattern and the anisotropic material properties into consideration. The optimized structure consists of a boundary layer and a substrate. The substrate domain is assumed to be filled with unidirectional zig-zag deposition paths and customized infill patterns, while the boundary is made by the contour offset deposition paths. This HDP is the most commonly employed path pattern for the fused deposition modeling (FDM) process. A critical derivative of the sensitivity analysis is presented in this paper, which ensures the optimality of the final design solutions. The effectiveness of the proposed method is validated through several 2D numerical examples.

## 1. Introduction

Additive manufacturing (AM) has gained fast development in research and industrial applications. The layer-by-layer material deposition nature of AM could enable graded material compositions and eliminate the design complexity constraints in conventional manufacturing methods [1]. AM breaks the restrictions between design and manufacturing and makes the greatest design freedom possible. Design for additive manufacturing (DfAM) has therefore attracted a great deal of attention [2,3].

Topology optimization has been widely treated as the main computational design method for AM [1,2,3,4], because it explores a large design space and successfully applies in different physical disciplines [5,6,7,8]. Diverse topology optimization methods have been proposed, including the homogenization method [9,10], solid isotropic material with penalization (SIMP) [11,12], evolutionary structural optimization (ESO) [13], the level set method (LSM) [14,15], the moving morphable component (MMC) method [16], and some others.

However, there are new challenges introduced by AM [2,17], especially for the fused deposition modeling (FDM) process [18,19]. As is widely recognized, AM produces anisotropic material properties whose mechanical performance in the raster direction, the transverse direction, and the build direction are evidently different [20,21,22]. It means that the deposition direction of the material significantly affects the structural performance. Focusing on this point, build direction has recently been explored as an optimization variable to improve structural performance [23,24,25,26,27]. In reality, most fused deposition modeling (FDM) machines only support the hybrid deposition path (HDP) pattern, i.e., external contour profile offsets through a constant distance T, where the deposition path is assumed to be distributing along the outer profile and the interior substrate structure is filled with fixed deposition paths of angle $\bar{\theta }$ (Figure 1). Under this scenario, to better close the practice, a level set-based method was proposed by Liu et al. to take the HDP pattern into account to address the material anisotropy topology optimization [28]. Besides, the field of fiber reinforcement composites is also highly relevant to additive manufacturing-oriented topology optimization [29,30]. An interdependent two-level optimization approach was proposed by Humberto et al. to optimize both fiber angle and intrinsic thickness, and producible results could be obtained by this method [31]. A novel framework for the optimized topology and the fiber paths was developed in [32]; the optimized distribution of the material and the fiber orientation are achieved by two methods: a density-based method and the level set method.

Currently, interface technology, like coating, is widely used in designing structures with enhanced functionalities and visual properties. To consider the interface issue from the view of structural optimization, other than simply designing the substrate structures, much research attention has been drawn to designing a mechanical system while considering the effect of the material interface by topology optimization. Clausen et al. [33] proposed a double smoothing and projection (DSP) approach to design the coated structure with an enhanced solid shell and a weak base structure. Luo et al. [34] developed an erosion-based method to design shell–infill structures. Based on the level set method, a topology optimization method, which considers bi-material coated structures, was proposed in [35]. Besides, a new density filter was developed by Yoon et al. [36] to conduct topology optimization, considering the coating structure. More currently, using moving morphable sandwich bars, an explicit topology optimization method for coated structures was developed in [37]. Note that the stress constrained interface problem was also investigated by Yu et al. [38].

Inspired by the interface problem, a new algorithm based on solid orthotropic material with penalization (SOMP) and DSP approaches is developed to find out an optimal structure design which considers the HDP pattern and the anisotropic material properties. In this work, the external offset contour (Figure 1) is assumed to be a uniform boundary layer structure which has a different material from the substrate structure. In particular, the DSP method allows the identification of the boundary layer and achieves the length control for it; the anisotropic material topology optimization could be realized by the SOMP interposition model.

The remainder of the paper is structured as follows: Section 2 presents the problem formulation. This includes the material model and the corresponding interpolation scheme, as well as the optimization problem and the sensitivity analysis. Section 3 presents the numerical implementation. Several numerical results are presented and discussed Section 4. Section 5 concludes the paper.

## 2. Problem Formulation

In this section, the optimization problem is defined. This includes defining an appropriate material model, formally defining the optimization problem, and deriving sensitivities. The material model and characteristic properties are derived analytically based on continuous versions of the design field and filters. The energy-based SOMP method is used to determine the structural topology; the DSP approach is adopted to distinguish the substrate structure and the boundary layer, whose implementation process is shown in Figure 2.

### 2.1. The Formulation for the Boundary Layer

According to the classical laminate theory, $D_{0}$ is the laminae unrotated compliance tensor [39,40]:(1)$$D_{0}=\left[\begin{array}{ccc} \frac{E_{x}}{1-v_{xy}v_{yx}} & \frac{v_{yx}E_{x}}{1-v_{xy}v_{yx}} & 0\\ \frac{v_{yx}E_{x}}{1-v_{xy}v_{yx}} & \frac{E_{y}}{1-v_{xy}v_{yx}} & 0\\ 0 & 0 & G_{xy} \end{array}\right],$$ and D(θ) is the 2D orthotropic elasticity tensor given any angle θ:(2)$$D\left(\theta \right)=T\left(\theta \right)D_{0}T{\left(\theta \right)}^{T},$$ with T(θ) being the transform matrix which is used to conduct the matrix coordinate transform:(3)$$T\left(\theta \right)=\left[\begin{array}{ccc} \cos^{2}\theta & \sin^{2}\theta & -2\sin\theta \cos\theta \\ \sin^{2}\theta & \cos^{2}\theta & 2\sin\theta \cos\theta \\ \sin\theta \cos\theta & -\sin\theta \cos\theta & \cos^{2}\theta -\sin^{2}\theta \end{array}\right]$$

In this work, the local fiber orientation θ could be analytically expressed by Equation (3) and counted in the counter-clockwise direction, as shown in Figure 3.
(4)$$\theta =\frac{\pi }{2}+arctan\left(\frac{\frac{\partial \hat{\varphi }}{\partial y}}{\frac{\partial \hat{\varphi }}{\partial x}}\right)$$
$\frac{\partial \hat{\varphi }}{\partial y}$ and $\frac{\partial \hat{\varphi }}{\partial x}$ are the spatial gradients of the filtered design field $\hat{\varphi }$, and their derivations could refer to later content.

Thus, the element stiffness matrix could be written as:(5)Ke=δ(D)
where δ(*) is the element stiffness matrix assembly operator.

### 2.2. The Optimization Model

In this paper, a standard compliance minimization problem, subject to mass constraint, is studied. The optimization problem is formulated as:(6)$$\{\begin{array}{c} Find:\;\mu_{E}\;\left(E=1,2,\ldots ,N\right)\\ \begin{array}{c} Minimize:C=U^{T}KU \end{array}\\ Subject\;to:\{\begin{array}{c} KU=F\\ G\leq M_{d}\\ 0<\mu_{min}\leq \mu_{E}\leq 1 \end{array} \end{array}$$
where $\mu_{E}$ is the design variable and the Eth element density and N is the total number of elements. K, U, and F are the global stiffness matrix, displacement vector, and force vector, respectively. $\mu_{min}$ is a small number to avoid matrix singularity. G is the mass fraction constrained by the maximum mass fraction $M_{d}$. Note that the detail expression of G could refer to later content.

### 2.3. Material Interpolation Strategy

Based on the SOMP, the element stiffness matrix and physical density interposition of the element E are defined as an interpolation of $\varphi_{E}$ and $\bar{{\left\|\nabla \hat{\varphi_{E}}\right\|}_{\alpha }}$:(7)$$\begin{array}{c} \begin{array}{c} K_{E}\left(\varphi_{E},\bar{{\left\|\nabla \hat{\varphi_{E}}\right\|}_{\alpha }}\right)={\left(\varphi_{E}\right)}^{p}\cdot Ke_{2}+{\left(\bar{{\left\|\nabla \hat{\varphi_{E}}\right\|}_{\alpha }}\right)}^{p}\cdot \bar{Ke_{1,E}}-{\left(\varphi_{E}\right)}^{p}\cdot {\left(\bar{{\left\|\nabla \hat{\varphi_{E}}\right\|}_{\alpha }}\right)}^{p}\cdot Ke_{2}\\ \rho_{E}\left(\varphi_{E},\bar{{\left\|\nabla \hat{\varphi_{E}}\right\|}_{\alpha }}\right)=\rho_{S}\cdot \varphi_{E}+\rho_{B}\cdot \left(1-\rho_{S}\cdot \varphi_{E}\right)\cdot {\left(\bar{{\left\|\nabla \hat{\varphi }\right\|}_{\alpha }}\right)}_{E} \end{array} \end{array}$$
where the $\bar{Ke_{1,E}}$ is the stiffness matrix of the mth element that belongs to the boundary layer, $Ke_{2}$ is the element stiffness matrix for the substrate material. p is employed to penalize the intermediate densities, so as to derive the black and white solution. Note that when $\bar{{\left\|\nabla \hat{\varphi_{E}}\right\|}_{\alpha }}$ approaches zero, i.e., when going away from the boundary layer, the expressions reduce to:(8)$$\begin{array}{c} \begin{array}{c} K_{E}\left(\varphi_{E},0\right)={\left(\varphi_{E}\right)}^{p}\cdot Ke_{2}\\ \rho_{E}\left(\varphi_{E},0\right)=\rho_{S}\cdot \varphi_{E} \end{array} \end{array}$$
where $\rho_{S}$ is the element density in the substrate domain.

Meanwhile, at the other extreme, when $\bar{{\left\|\nabla \hat{\varphi_{E}}\right\|}_{\alpha }}$ approaches 1, i.e., at the boundary layer region, the expressions could be expressed as:(9)$$\begin{array}{c} \begin{array}{c} K_{E}\left(\varphi_{E},1\right)=\bar{Ke_{1,E}}\\ \rho_{E}\left(\varphi_{E},1\right)=\rho_{B} \end{array} \end{array}$$
where $\rho_{B}$ is the element density in the boundary area.

In conclusion, both the physical density and stiffness are interpolated based on the projected design variable field, $\varphi_{E}$, and the normalized gradient of the filtered field $\bar{{\left\|\nabla \hat{\varphi_{E}}\right\|}_{\alpha }}$.

### 2.4. Objective Function

The structural compliance is equal to the sum of the element strain energy, which could be expressed as:(10)$$C=\overset{N}{\underset{E=1}{\sum }}Q_{E}$$
where $Q_{E}$ is the strain energy of the element E:(11)$$Q_{E}=U_{E}{}^{T}K_{E}U_{E}$$

Substituting the element stiffness matrix interpolation in Equation (7) into Equation (11) will yield:(12)$$\begin{array}{c} Q_{E}={\left(\varphi_{E}\right)}^{p}\cdot \left(U_{E}{}^{T}Ke_{2}U_{E}\right)+{\left(\bar{{\left\|\nabla \hat{\varphi_{E}}\right\|}_{\alpha }}\right)}^{p}\cdot \left(U_{E}{}^{T}\bar{Ke_{1,E}}U_{E}\right)\\ -{\left(\varphi_{E}\right)}^{p}\cdot {\left(\bar{{\left\|\nabla \hat{\varphi }\right\|}_{\alpha }}\right)}_{E}{}^{p}\cdot \left(U_{E}{}^{T}Ke_{2}U_{E}\right) \end{array}$$


For brevity, the following notation is introduced:
(13)$$Q_{E}=\omega_{S}+\omega_{G}+\omega_{SG}=\varepsilon_{E}^{S}\cdot Q2_{E}+\varepsilon_{E}^{G}\cdot Q1_{E}-\varepsilon_{E}^{SG}\cdot Q2_{E}$$
where $\varepsilon_{E}^{S}$, $\varepsilon_{E}^{G}$, and $\varepsilon_{E}^{SG}$ could be treated as three penalized pseudo-density fields:(14)$$\{\begin{array}{c} \varepsilon_{E}^{S}={\left(\varphi_{E}\right)}^{p}\\ \varepsilon_{E}^{G}={\left(\bar{{\left\|\nabla \hat{\varphi_{E}}\right\|}_{\alpha }}\right)}^{p}\\ \varepsilon_{E}^{SG}={\left(\varphi_{E}\right)}^{p}\cdot {\left(\bar{{\left\|\nabla \hat{\varphi_{E}}\right\|}_{\alpha }}\right)}^{p} \end{array}$$

$Q2_{E}$ is the element strain energy for the substrate element and $Q1_{E}$ is treated as the modified element strain energy for the boundary layer element:(15)$$\{\begin{array}{c} Q1_{E}=U_{E}{}^{T}\bar{Ke_{1,E}}U_{E}\\ Q2_{E}=U_{E}{}^{T}Ke_{2}U_{E} \end{array}$$

### 2.5. Mass Constraint

In this paper, the mass constraint function could be written as:(16)$$G=\overset{N}{\underset{E=1}{\sum }}G_{E}$$
where $G_{E}$ is the mass of the element E:(17)$$G_{E}=M_{0}\cdot \rho_{E}=M_{0}\cdot \left[\rho_{S}\cdot \varphi_{E}+\rho_{B}\cdot \left(1-\rho_{S}\cdot \varphi_{E}\right)\cdot {\left(\bar{{\left\|\nabla \hat{\varphi }\right\|}_{\alpha }}\right)}_{E}\right]$$

It is evident that the macroscale volume constraint considers both the substrate and coating materials. Like the notation form used in the last subsection, the following expression is introduced:(18)$$G_{E}=M_{0}\cdot \left(\epsilon_{E}^{S}+\epsilon_{E}^{G}-\epsilon_{E}^{SG}\right)$$
where $M_{0}$ is the design element standard mass; $\epsilon_{E}^{S}$, $\epsilon_{E}^{G}$, and $\epsilon_{E}^{SG}$ are the three non-penalized pseudo-design fields, as follows:(19)$$\{\begin{array}{c} \epsilon_{E}^{S}=\rho_{S}\cdot \varphi_{E}\\ \epsilon_{E}^{G}=\rho_{B}\cdot \bar{{\left\|\nabla \hat{\varphi_{E}}\right\|}_{\alpha }}\\ \epsilon_{E}^{SG}=\rho_{S}\cdot \rho_{B}\cdot \varphi_{E}\cdot \bar{{\left\|\nabla \hat{\varphi_{E}}\right\|}_{\alpha }} \end{array}$$

### 2.6. Sensitivity Analysis

The updates of the design variables are performed based on sensitivity analysis using the Method of Moving Asymptote (MMA) algorithm, which requires first order sensitivity information of the constraints and the objective function. In this subsection, a critical derivative of the sensitivity analysis is presented.

#### 2.6.1. Sensitivity Analysis for Objective Function

Recalling Equation (13), $\frac{\partial Q_{E}}{\partial \mu_{E}}$ could be written in the following form:(20)$$\frac{\partial Q_{E}}{\partial \mu_{E}}=\frac{\partial \omega_{S}}{\partial \mu_{E}}+\frac{\partial \omega_{G}}{\partial \mu_{E}}+\frac{\partial \omega_{SG}}{\partial \mu_{E}}$$

The first derivative term could be obtained using the chain rule:(21)$$\frac{\partial \omega_{S}}{\partial \mu_{E}}=\frac{\partial \varepsilon_{E}^{S}}{\partial \mu_{E}}\cdot Q2_{E}=\frac{\partial {\left(\varphi_{E}\right)}^{p}}{\partial \mu_{E}}\cdot Q2_{E}=p\cdot \varphi_{E}{}^{p-1}\cdot \frac{\partial \varphi_{E}}{\partial {\hat{\mu }}_{E}}\cdot \frac{\partial \hat{\mu_{E}}}{\partial \mu_{E}}\cdot Q2_{E}$$

The term $\frac{\partial \varphi_{E}}{\partial {\hat{\mu }}_{E}}$ represents the standard modification of sensitivities due to projection and the detail expression can be found in [41]. $\frac{\partial \hat{\mu_{E}}}{\partial \mu_{E}}$ is the standard modification of the smoothing filter and its derivative will be discussed in a later subsection.

The second derivative term of Equation (20) is given by:(22)$$\frac{\partial \omega_{G}}{\partial \mu_{E}}=\frac{\partial \varepsilon_{E}^{G}}{\partial \mu_{E}}\cdot Q1_{E}+\frac{\partial Q1_{E}}{\partial \mu_{E}}\cdot \varepsilon_{E}^{G}$$
and the first derivative term could be written as the following form:(23)$$\frac{\partial \varepsilon_{E}^{G}}{\partial \mu_{E}}=\frac{\partial \varepsilon_{E}^{G}}{\partial {\hat{\varphi }}_{N}}\cdot \frac{\partial {\hat{\varphi }}_{N}}{\partial \varphi_{N}}\cdot \frac{\partial \varphi_{N}}{\partial \varphi_{E}}\cdot \frac{\partial \varphi_{E}}{\partial {\hat{\mu }}_{E}}\cdot \frac{\partial {\hat{\mu }}_{E}}{\partial \mu_{E}}$$
where
(24)$$\frac{\partial \varepsilon_{E}^{G}}{\partial {\hat{\varphi }}_{N}}=p\cdot {\left(\bar{{\left\|\nabla \hat{\varphi_{E}}\right\|}_{\alpha }}\right)}^{p-1}\cdot \frac{\partial \left(\bar{{\left\|\nabla \hat{\varphi_{E}}\right\|}_{\alpha }}\right)}{\partial {\hat{\varphi }}_{N}}=p\cdot {\left(\bar{{\left\|\nabla \hat{\varphi_{E}}\right\|}_{\alpha }}\right)}^{p-1}\cdot \frac{\partial \left(\bar{{\left\|\nabla \hat{\varphi_{E}}\right\|}_{\alpha }}\right)}{\partial \left({\left\|\nabla \hat{\varphi_{E}}\right\|}_{\alpha }\right)}\cdot \frac{\partial \left({\left\|\nabla \hat{\varphi_{E}}\right\|}_{\alpha }\right)}{\partial {\hat{\varphi }}_{N}}$$
in which the term $\frac{\partial \left(\bar{{\left\|\nabla \hat{\varphi_{E}}\right\|}_{\alpha }}\right)}{\partial \left({\left\|\nabla \hat{\varphi_{E}}\right\|}_{\alpha }\right)}$ indicates the standard modification of sensitivities due to the projection of the ${\left\|\nabla \hat{\varphi_{E}}\right\|}_{\alpha }$ field. The derivative of the normalized gradient norm $\frac{\partial \left({\left\|\nabla \hat{\varphi_{E}}\right\|}_{\alpha }\right)}{\partial {\hat{\varphi }}_{N}}$ could be written as:(25)$$\frac{\partial \left({\left\|\nabla \hat{\varphi_{E}}\right\|}_{\alpha }\right)}{\partial {\hat{\varphi }}_{N}}=\frac{\alpha }{\left\|\nabla \hat{\varphi }\right\|}\cdot \left(\frac{\partial {\hat{\varphi }}_{E}}{\partial x}\frac{\partial \frac{\partial {\hat{\varphi }}_{E}}{\partial x}}{\partial {\hat{\varphi }}_{N}}+\frac{\partial {\hat{\varphi }}_{E}}{\partial y}\frac{\partial \frac{\partial {\hat{\varphi }}_{E}}{\partial y}}{\partial {\hat{\varphi }}_{N}}\right)$$
in which:(26)$$\begin{array}{c} \frac{\partial {\hat{\varphi }}_{E}}{\partial x}=\frac{\partial \left(N^{T}{\hat{\varphi }}_{N}\right)}{\partial x}=B_{x}{\hat{\varphi }}_{N}\\ \frac{\partial {\hat{\varphi }}_{E}}{\partial y}=\frac{\partial \left(N^{T}{\hat{\varphi }}_{N}\right)}{\partial y}=B_{y}{\hat{\varphi }}_{N} \end{array}$$
where N is a vector of the four shape functions relating nodal variable ${\hat{\varphi }}_{N}$ with the elemental variable ${\hat{\varphi }}_{E}$; $B_{x}$ and $B_{y}$ are the gradient computation matrices for N in the x and y directions, respectively, and are independent from the design variables. Therefore, we have:(27)$$\begin{array}{c} \frac{\partial \frac{\partial {\hat{\varphi }}_{E}}{\partial x}}{\partial {\hat{\varphi }}_{N}}=\frac{\partial \left(B_{x}{\hat{\varphi }}_{N}\right)}{\partial {\hat{\varphi }}_{N}}=B_{x}\\ \frac{\partial \frac{\partial {\hat{\varphi }}_{E}}{\partial y}}{\partial {\hat{\varphi }}_{N}}=\frac{\partial \left(B_{y}{\hat{\varphi }}_{N}\right)}{\partial {\hat{\varphi }}_{N}}=B_{y} \end{array}$$

Thus,
(28)$$\frac{\partial \varepsilon_{E}^{G}}{\partial {\hat{\varphi }}_{N}}=p\cdot {\left(\bar{{\left\|\nabla \hat{\varphi_{E}}\right\|}_{\alpha }}\right)}^{p-1}\cdot \frac{\partial \left(\bar{{\left\|\nabla \hat{\varphi_{E}}\right\|}_{\alpha }}\right)}{\partial \left({\left\|\nabla \hat{\varphi_{E}}\right\|}_{\alpha }\right)}\cdot \frac{\alpha }{\left\|\nabla \hat{\varphi }\right\|}\cdot \left(\frac{\partial {\hat{\varphi }}_{E}}{\partial x}B_{x}+\frac{\partial {\hat{\varphi }}_{E}}{\partial y}B_{y}\right)$$
and the following notation of Equation (28) is introduced:(29)$$\frac{\partial \varepsilon_{E}^{G}}{\partial {\hat{\varphi }}_{N}}=M_{E}{}^{x}B_{x}+M_{E}{}^{y}B_{y}$$
where
(30)$$\begin{array}{c} M_{E}{}^{x}=p\cdot {\left(\bar{{\left\|\nabla \hat{\varphi_{E}}\right\|}_{\alpha }}\right)}^{p-1}\cdot \frac{\partial \left(\bar{{\left\|\nabla \hat{\varphi_{E}}\right\|}_{\alpha }}\right)}{\partial \left({\left\|\nabla \hat{\varphi_{E}}\right\|}_{\alpha }\right)}\cdot \frac{\alpha }{\left\|\nabla \hat{\varphi }\right\|}\cdot \frac{\partial {\hat{\varphi }}_{E}}{\partial x}\\ M_{E}{}^{y}=p\cdot {\left(\bar{{\left\|\nabla \hat{\varphi_{E}}\right\|}_{\alpha }}\right)}^{p-1}\cdot \frac{\partial \left(\bar{{\left\|\nabla \hat{\varphi_{E}}\right\|}_{\alpha }}\right)}{\partial \left({\left\|\nabla \hat{\varphi_{E}}\right\|}_{\alpha }\right)}\cdot \frac{\alpha }{\left\|\nabla \hat{\varphi }\right\|}\cdot \frac{\partial {\hat{\varphi }}_{E}}{\partial y} \end{array}$$

For the term $\frac{\partial Q1_{E}}{\partial \mu_{E}}$, it could have following form:(31)$$\frac{\partial Q1_{E}}{\partial \mu_{E}}=\frac{\partial Q1_{E}}{\partial \theta_{E}}\cdot \frac{\partial \theta_{E}}{\partial {\hat{\varphi }}_{N}}\cdot \frac{\partial {\hat{\varphi }}_{N}}{\partial \varphi_{N}}\cdot \frac{\partial \varphi_{N}}{\partial \varphi_{E}}\cdot \frac{\partial \varphi_{E}}{\partial {\hat{\mu }}_{E}}\cdot \frac{\partial {\hat{\mu }}_{E}}{\partial \mu_{E}}$$
where
(32)$$\frac{\partial Q1_{E}}{\partial \theta_{E}}=\frac{\partial \left(U_{E}{}^{T}Ke_{1,E}U_{E}\right)}{\partial \theta_{E}}=U_{E}{}^{T}\delta \left(\frac{\partial D_{E}}{\partial \theta_{E}}\right)U_{E}$$

Substituting the material interpolation of Equation (2) into the term $\frac{\partial D_{E}}{\partial \theta_{E}}$ will yield:(33)$$\frac{\partial D_{E}}{\partial \theta_{E}}=U_{E}{}^{T}\delta \left(\frac{\partial \left(T\left(\theta \right)D_{0}T{\left(\theta \right)}^{T}\right)}{\partial \theta_{E}}\right)U_{E}$$

The elastic tensor of matrix material $D_{0}$ is independent of $\mu_{E}$. Using the chain rule again, we could arrive at:(34)$$\frac{\partial \left(T\left(\theta \right)D_{0}T{\left(\theta \right)}^{T}\right)}{\partial \theta_{E}}=\frac{\partial T\left(\theta_{E}\right)}{\partial \theta_{E}}D_{0}T{\left(\theta_{E}\right)}^{T}+T\left(\theta_{E}\right)D_{0}\frac{\partial T{\left(\theta_{E}\right)}^{T}}{\partial \theta_{E}}$$
where
(35)$$\frac{\partial T\left(\theta_{E}\right)}{\partial \theta_{E}}=\left[\begin{array}{ccc} -2\cdot \sin\theta \cdot \cos\theta & \sin2\theta & -2\cdot \cos2\theta \\ \sin2\theta & -2\sin\theta \cdot \cos\theta & 2\cdot \cos2\theta \\ \cos2\theta & -\cos2\theta & -2\cdot \sin\theta \cdot \cos\theta -\sin2\theta \end{array}\right]$$

Then, the derivative of $\frac{\partial \theta_{E}}{\partial {\hat{\varphi }}_{N}}$ is obtained as:(36)$$\frac{\partial \theta_{E}}{\partial {\hat{\varphi }}_{N}}=\frac{1}{1+{\left(\frac{\partial {\hat{\varphi }}_{E}}{\partial y}\right)}^{2}\cdot {\left(\frac{\partial {\hat{\varphi }}_{E}}{\partial x}\right)}^{-2}}\cdot \frac{\frac{\partial {\hat{\varphi }}_{E}}{\partial x}\cdot B_{y}-\frac{\partial {\hat{\varphi }}_{E}}{\partial y}\cdot B_{x}}{{\left(\frac{\partial {\hat{\varphi }}_{E}}{\partial x}\right)}^{2}}$$

Similarly, the following notation of Equation (36) is introduced:(37)$$\frac{\partial \theta_{E}}{\partial {\hat{\varphi }}_{N}}={{\bar{M}}_{E}}^{x}B_{y}+{{\bar{M}}_{E}}^{y}B_{x}$$
where
(38)$$\begin{array}{c} {{\bar{M}}_{E}}^{x}=\frac{1}{1+{\left(\frac{\partial {\hat{\varphi }}_{E}}{\partial y}\right)}^{2}\cdot {\left(\frac{\partial {\hat{\varphi }}_{E}}{\partial x}\right)}^{-2}}\cdot \frac{1}{\frac{\partial {\hat{\varphi }}_{E}}{\partial x}}\\ {{\bar{M}}_{E}}^{y}=\frac{1}{1+{\left(\frac{\partial {\hat{\varphi }}_{E}}{\partial y}\right)}^{2}\cdot {\left(\frac{\partial {\hat{\varphi }}_{E}}{\partial x}\right)}^{-2}}\cdot \frac{\frac{\partial {\hat{\varphi }}_{E}}{\partial y}}{{\left(\frac{\partial {\hat{\varphi }}_{E}}{\partial x}\right)}^{2}} \end{array}$$

The third derivative term of Equation (20) could be obtained simply by using the product rule and all the terms already obtained above:(39)$$\frac{\partial \varepsilon_{E}^{SG}}{\partial \mu_{E}}\cdot Q2_{E}=\frac{\partial \varepsilon_{E}^{S}}{\partial \mu_{E}}\cdot \varepsilon_{E}^{G}\cdot Q2_{E}+\frac{\partial \varepsilon_{E}^{G}}{\partial \mu_{E}}\cdot \varepsilon_{E}^{S}\cdot Q2_{E}$$

#### 2.6.2. Sensitivity Analysis for Mass Constraint

For the term $\frac{\partial G_{E}}{\partial \mu_{E}}$, it could be similarly elaborated by the following expressions:(40)$$\frac{\partial G_{E}}{\partial \mu_{E}}=M_{0}\cdot \frac{\partial \rho_{E}}{\partial \mu_{E}}=M_{0}\cdot \frac{\partial \left(\epsilon_{E}^{S}+\epsilon_{E}^{G}-\epsilon_{E}^{SG}\right)}{\partial \mu_{E}}$$
where,
(41)$$\{\begin{array}{c} \frac{\partial \epsilon_{E}^{S}}{\partial \mu_{E}}=\rho_{S}\cdot \frac{\partial \varphi_{E}}{\partial \hat{\mu_{E}}}\cdot \frac{\partial \hat{\mu_{E}}}{\partial \mu_{E}}\\ \frac{\partial \epsilon_{E}^{G}}{\partial \mu_{E}}=\rho_{B}\cdot \frac{\partial \bar{{\left\|\nabla \hat{\varphi_{E}}\right\|}_{\alpha }}}{\partial \mu_{E}}\\ \frac{\partial \epsilon_{E}^{SG}}{\partial \mu_{E}}=\rho_{S}\cdot \rho_{B}\cdot \left(\frac{\partial \epsilon_{E}^{S}}{\partial \mu_{E}}\cdot \epsilon_{E}^{G}+\frac{\partial \epsilon_{E}^{G}}{\partial \mu_{E}}\cdot \epsilon_{E}^{S}\right) \end{array}$$

All the derivative terms in Equation (41) could be obtained by the previous expressions.

#### 2.6.3. Filtering Based on Helmholtz-Type Differential Equations

The smoothing filter adopted in this work could be implicitly represented by the solution of a Helmholtz-type partial differential equation (PDE) [42]:(42)$$-R^{2}\nabla^{2}\hat{\mu }+\hat{\mu }=\mu$$
by imposing the homogeneous Neumann boundary conditions ($\frac{\partial \hat{\mu }}{\partial n}=0$) on the boundary of the design domain. The solution of Equation (42) can be written in a convolution integral form, which has a similar function to the classical filter [43]. In Equation (42), μ represents the unfiltered design field, and $\hat{\mu }$ is the filtered field; the parameter R plays a similar role as the minimum filter radius ($r_{min}$) in the classical filter [43]. An approximate relation between the length scales for the classical filter and the PDE filter is given by [42]:(43)$$R=\frac{r_{min}}{2\sqrt{3}}$$

Using the finite element method to discrete Equation (42):(44)$$K_{x}{\hat{\mu }}_{N}=\mu_{N}$$
where $K_{x}$ is the standard stiffness matrix in finite element method for the scalar problem corresponding to $R_{x}$ and ${\hat{\mu }}_{N}$ is the representation of the filtered nodal field. Thus, the derivative of the filtered sensitives, with respect to the design variable, can be written as:(45)$$\frac{\partial {\hat{\mu }}_{N}}{\partial \mu_{N}}=K_{x}{}^{-1}$$

The element node representation of the field is obtained by:(46)$$\mu_{N}=T_{F}\mu$$
where $T_{F}$ is a matrix which maps the elemental values μ to a vector with nodal values $\mu_{N}$. Similarly, the element node representation of the filtered field could be expressed as:(47)$$\hat{\mu }=T_{F}{}^{T}{\hat{\mu }}_{N}$$
and
(48)$$\frac{\partial \mu_{N}}{\partial \mu }=T_{F}$$

According to the above derivation, the filtered sensitivities of the term $\omega_{S}$ can be calculated as:(49)$$\frac{\partial \omega_{S}}{\partial \mu }=T_{F}{}^{T}\left(K_{1}{}^{-1}\left(T_{F}\left(p\cdot \varphi^{p-1}\cdot \frac{\partial \varphi }{\partial \hat{\mu }}\cdot Q2\right)\right)\right)$$
the filtered sensitivities of the term $\omega_{G}$ could be rewritten as:(50)$$\frac{\partial \omega_{G}}{\partial \mu }=\frac{\partial \omega_{G1}}{\partial \mu }+\frac{\partial \omega_{G2}}{\partial \mu }$$

The terms $\frac{\partial \omega_{G1}}{\partial \mu }$ and $\frac{\partial \omega_{G2}}{\partial \mu }$ can be computed as:(51)$$\begin{array}{c} \frac{\partial \omega_{G1}}{\partial \mu }=T_{F}{}^{T}\left(K_{1}{}^{-1}\left(T_{F}\left(dc\cdot \frac{\partial \varphi }{\partial \hat{\mu }}\right)\right)\right)\\ dc=T_{F}{}^{T}\left(K_{2}{}^{-1}\left(T_{F}{\left(M^{x}\cdot Q1\right)}^{T}B_{X}{}^{T}+{\left(M^{y}\cdot Q1\right)}^{T}B_{Y}{}^{T}\right)\right) \end{array}$$
and
(52)$$\begin{array}{c} \frac{\partial \omega_{G2}}{\partial \mu }=T_{F}{}^{T}\left(K_{1}{}^{-1}\left(T_{F}\left(\bar{dc}\cdot \frac{\partial \varphi }{\partial \hat{\mu }}\right)\right)\right)\\ \bar{dc}=T_{F}{}^{T}\left(K_{2}{}^{-1}\left(T_{F}\left({\left({\bar{M}}^{x}\cdot \varepsilon^{G}\right)}^{T}B_{X}{}^{T}+{\left({\bar{M}}^{y}\cdot \varepsilon^{G}\right)}^{T}B_{Y}{}^{T}\right)\right)\right) \end{array}$$

Again, the filtered derivative for the term $\omega_{SG}$ and the filtered sensitivity analysis for the mass constraint could be obtained in a similar way and are therefore omitted here.

## 3. Numerical Implementations

The proposed method is validated with several classical 2D benchmark cases in the next section. Four-node quadrilateral elements are adopted in all numerical examples. For the MMA optimizer, the default move limit is 0.3. Additionally, following the suggestion in Ref. [33], a continuation strategy for projection is adopted, where the sharpness factor for the substrate projection is set as $\beta_{S}=1$ at the beginning of optimization and gradually increased to 64 by doubling every 50 iterations (or at convergence), while the sharpness factor for the boundary layer projection is initialized with $\beta_{G}=4$ to ensure a sharp coating from the first iteration, and doubled every 50 iterations (or at convergence) until it is increased to 128. A projection threshold of 0.5 is used for $\mu_{S}$ and $\mu_{G}$. The iterative process terminates when no further improvement in the objective function can be achieved, namely, when the difference in the objective values between two adjacent iterations is less than 0.01 or the maximum iterative number is exceeded. The whole process of the proposed method is shown in Figure 4.

## 4. Case Studies

### 4.1. Messerschmidt–Bölkow–Blohm (MBB) Problem

#### 4.1.1. The Fully Infilled Substate Problem

The MBB beam problem is investigated to minimize the structural compliance under the maximum material volume ratio of 0.5, whose boundary condition is shown in Figure 5. The structural sizes are defined with L=30 and H=10. Only one half of the structure is optimized due to the symmetry condition. The MBB structure is loaded with a concentrated vertical force (F=1) at the upper left corner; the bottom right corner is supported on a roller; and the asymmetrical boundary condition is applied to the left edge. The nodal displacement in the x-direction is restricted, while in the y-direction it is free.

A solid material with a Young’s Modulus of 2.0 GPa in the raster direction and 0.5 GPa in the transverse direction is used. In addition, the Poisson’s ratio is 0.4, and the shear modulus is 0.35 GPa. The substrate material is assumed to be fully infilled ($\rho_{S}=\rho_{B}$) and the direction and the rotation angle θ of the raster direction is defined positively in the counter-clockwise direction (which is consistent with the depiction in Figure 3).

The boundary layer width T=4 ($R_{1}=14$ and $R_{2}=10$) is investigated in the first test. The raster direction for the substrate material is assumed to be 0°. In order to get a clear-cut solid structural design within the solid area, $\rho_{E}\geq 0.95$ indicates a clearly formed substrate domain, and $\bar{{\left\|\nabla \hat{\varphi_{E}}\right\|}_{\alpha }}\geq 0.5$ represents a clearly formed boundary layer. The final compliance is 88.3850, and the optimization terminates at the 320th iteration. The optimized result is shown in Figure 6.

From the convergence history graphs (Figure 7), it is seen that several sudden changes happen after the update of $\beta_{S}$ and $\beta_{G}$, but it gradually becomes stable and finally converges.

The detailed evolution of the topology of the substrate domain and boundary layer for the case in Figure 6 is given in Figure 8. As can be observed, the approximated structural topology is formed before the 200th iteration, and a clearer substate domain and boundary could be given in the last 120 iterations. Note that the boundary layer initially appears at the top left and bottom right corners, because the boundary layer (or solid substrate material) is required at all loads and helps to overrule the zero Dirichlet condition for the PDE filter [33].

#### 4.1.2. Comparing with the Result from the Non-Boundary Layer Structure

In comparison, the structure without the boundary layer, under the same condition, is optimized in this test, and its optimized result is depicted in Figure 9. The raster direction of the substrate material is defined as 0°. To be specific, the filter radius ($R_{2}=15$) is consistent with the case in Figure 6. Again, within the solid area, $\rho_{E}\geq 0.95$ indicates a clearly formed domain. It is clear to see that, under the same raster direction, the optimization result with the boundary layer (c=88.3850) is better than the one without the boundary layer (c=100.6204), and the compliance performance is 12.16% smaller than the latter one.

#### 4.1.3. The Influence of Different Raster Directions

In order to investigate the influence of different raster directions, this example explores the topology optimization with three designable raster directions (starting from 0°, 90°, and 45°) with the same boundary layer width (T=4) and boundary condition. Correspondingly, the optimization results are demonstrated in Figure 10.

The optimization results with the raster directions of 0°, 90°, and 45° have distinct structural topologies and shapes, and their structural performance is also different. The optimization result with the raster direction of 0° is better than the ones with the raster directions of 45° and 90°. This is reasonable from a mechanics point of view in sense that the principal stresses are distributed along the horizontal direction of the beam in the presented MBB problem.

#### 4.1.4. The Influence of Different Boundary Layer Widths

In Figure 11, the same design problem as above is solved with the raster direction θ=0° under the same conditions and varying boundary layer widths (T=2, T=4 and T=6). The modeled boundary layer width is clearly controlled by modifying the filter radius $R_{2}$, and the compliance improves when increasing the boundary layer width. In order to assure sufficiently wide features in the base structure, the first smoothing radius $R_{1}$ should be greater than or equal to the second smoothing radius $R_{2}$ [33]. Note that, in this work, the length control function could be implicitly achieved through the application of smoothing and projection filters. Increasing $R_{1}$ will lead to an increase in the minimum feature length and thereby eliminate some small geometry features.

#### 4.1.5. The Mesh Independence

Figure 12 shows the optimized structure discretized by three different element side lengths: 0.05, 0.1, and 0.2, respectively. The same topology and similar shapes could be found in the three final designs. The thickness of the boundary layer is almost independent of mesh size and highly uniform.

### 4.2. Cantilever Problem

#### 4.2.1. The Fully Infilled Substrate Problem

Next, the optimization of the cantilever problem is conducted. The boundary conditions are presented in Figure 13; its left side is clamped and the middle point of the right side is loaded with a constant force (F=1 ). The structural sizes are defined by L=30 and H=15. The mesh is discretized by 200×100 elements. The maximum material volume ratio is 0.5 in this case. 

Three cases with fully infilled substrate materials under different raster directions (0°, 45°, and 90°) are given in Figure 14.

#### 4.2.2. The Thick Boundary Problem

Finally, the structure with a thick boundary layer under the raster direction of 0° is investigated. The optimized objective value is 32.0089, which is lower than the result in Figure 14a. In order to guarantee a constant boundary layer thickness for the optimized structure, a relatively high minimum feature size is needed, and the boundary layer thickness should be much smaller than the feature size for the substrate structure. Besides, within our thick width results (Figure 15), we find the desired interface width dictated by the mesh resolution. For example, under the coarse mesh resolution (100×50) adopted in the same case in Figure 15, it is impossible to identify a boundary layer that is equal or above T=10.

Figure 16 shows the boundary layer raster direction distribution. The arrow indicates the raster direction for each boundary element. As can be seen in Figure 16, it is less prone to sudden orientation changes in the boundary layer domain, and it is matched well with the boundary layer structure.

### 4.3. Short Cantilever Problem

#### 4.3.1. The Fully Infilled Substrate Problem

The third numerical test is a short cantilever beam illustrated in Figure 17, where the left side is clamped and the middle point of the right side is loaded with a constant force (F=1 ). The structural sizes are defined as L=15 and H=30. The whole design domain is meshed by 100×200 elements. 

Firstly, four fully infilled substrate materials with different raster directions (0°, 90°, and 45°) under a mass fraction constraint of 0.25 are considered in this subsection, and their optimized results are shown in Figure 18.

#### 4.3.2. The Customized Infilled Pattern Problem

In this subsection, four customized infill patterns (wiggle, honeycomb, and two line infills) with 64.75%, 53.52%, 79.21%, and 79.10% density, respectively, are considered in this subsection. Their effective elastic tensors are predicted through the energy-based homogenization method [44,45], and the detail geometry structure, raster direction distribution, and effective elastic tensors are demonstrated in Figure 19. The problem configuration and the material properties are the same as the previous example, except the boundary layer width is T=3.

The results are demonstrated in Figure 20, and the derived objective values are 41.4360, 14.6845, 27.5177, and 28.1572 for wiggle, honeycomb, 45° line, and −45° line, respectively.

Finally, the computing time is briefly discussed. All the above cases were run on a desktop computer with Intel Xeon W-2145 CPU and 64GB RAM. Then, for the cases with the mesh dimension 300×100, an average of 18s is taken for each iteration: the FEM part takes 63.16%, the sensitivity analysis takes 18.42%, the MMA solver takes 15.79%, and the other parts take 2.63%. Meanwhile, for the cases with 200×100 elements, the algorithm takes 13s for each iteration on average: the FEM part takes 56.92%, the sensitivity analysis takes 17.12%, the MMA solver takes 19.23%, and the other parts take 6.73%. We could conclude that the FEM part takes more than half of the time in this method, and its computational cost grows rapidly with an increase in the dimension of the mesh. Therefore, a high efficiency FEM solver is demanded in this work.

## 5. Conclusions

The HDP pattern could be supported by most commercial tool path planning toolkits. Therefore, the HDP-based structure optimization could get closer to practice. Compared with the work proposed in [28] under the level set framework, based on solid SOMP and DSP approaches, this work proposes a methodology to find an optimal structure design, which takes the HDP pattern and the anisotropic material properties into consideration. The HDP pattern optimization here is assumed to be a structure optimization problem including coated structures, and the anisotropic material topology optimization is achieved by SOMP. The effectiveness of the proposed method are proven by several case studies, and the influence of different substrate raster directions under different boundary layer thicknesses is investigated. Note that the hybrid deposition paths produced in this work only provide the pattern where the zig-zag domain plays a significant role.

However, a unidirectional zig-zag deposition path is defined inside the substrate domain for the sake of simplicity. In fact, an optimized deposition path could achieve an even better design performance. The authors also intend to extend the proposed methods to address 3D problems. Besides, experimental validation is a must. These aspects will be explored in our future work as well.

## Figures and Tables

**Figure 1 micromachines-11-00709-f001:**
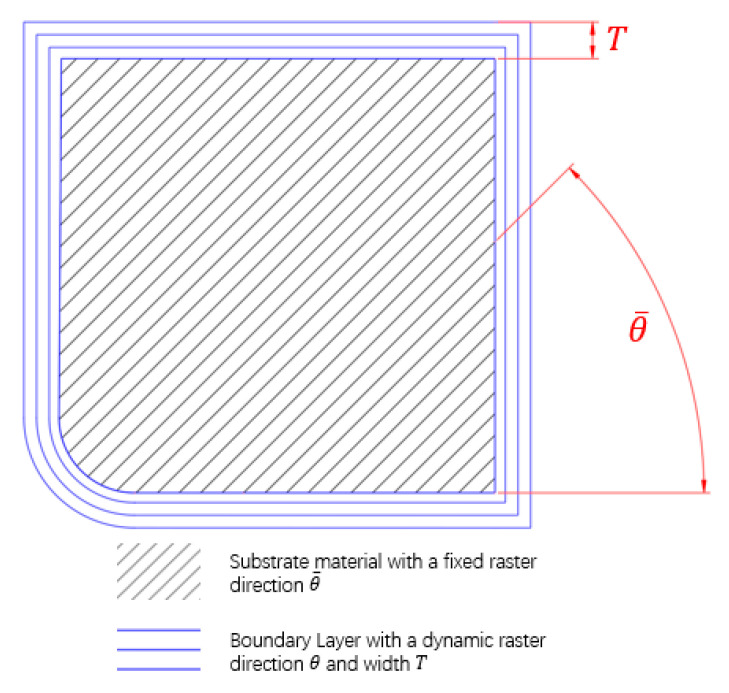
The hybrid deposition paths.

**Figure 2 micromachines-11-00709-f002:**
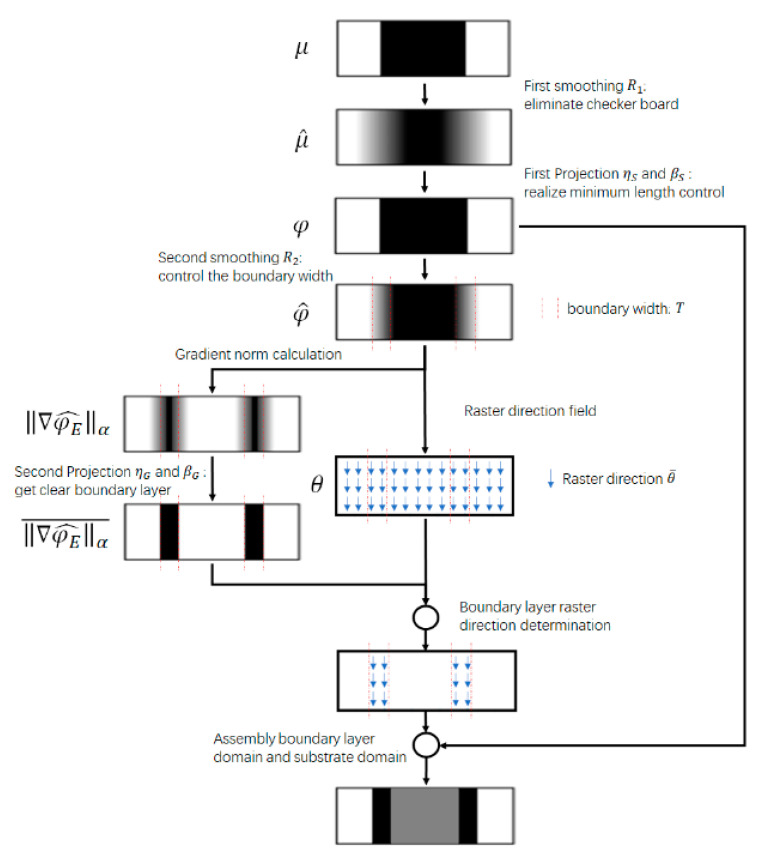
The procedure of the boundary layer deposition direction determination.

**Figure 3 micromachines-11-00709-f003:**
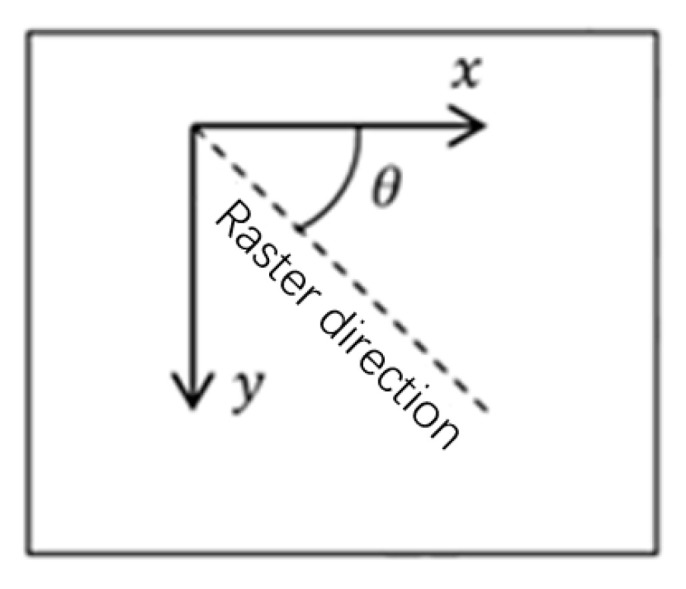
The axes and the rotation angle.

**Figure 4 micromachines-11-00709-f004:**
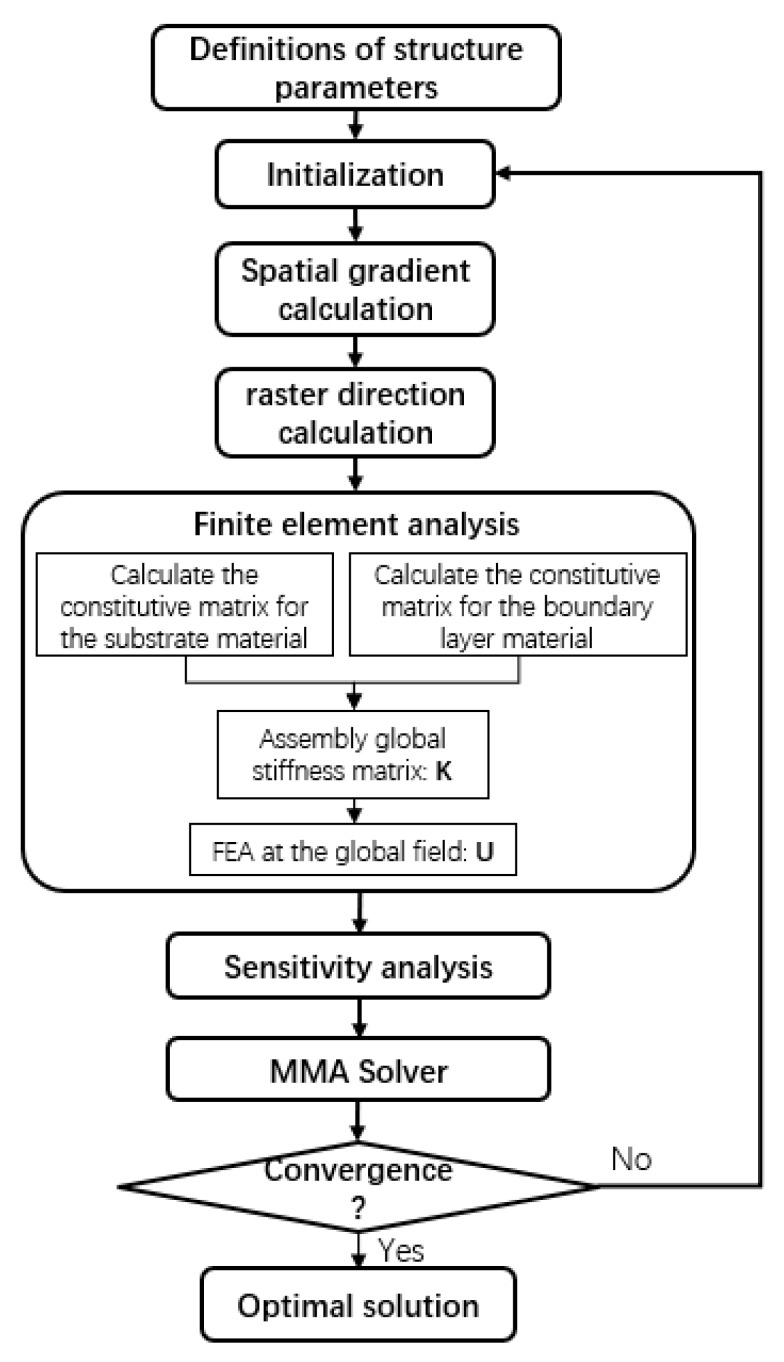
The flow chart of the proposed method.

**Figure 5 micromachines-11-00709-f005:**
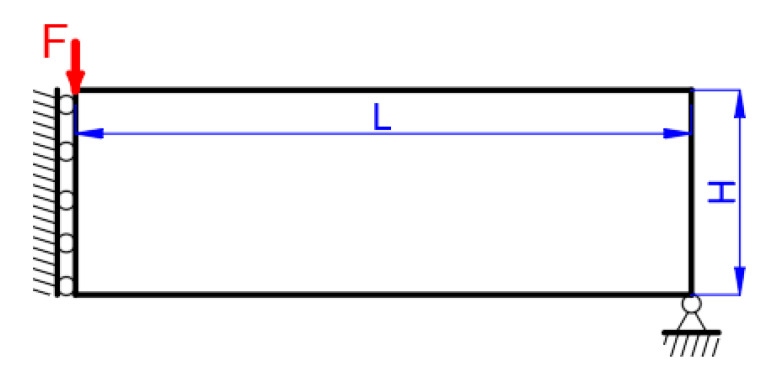
The Messerschmidt–Bölkow–Blohm (MBB) beam.

**Figure 6 micromachines-11-00709-f006:**
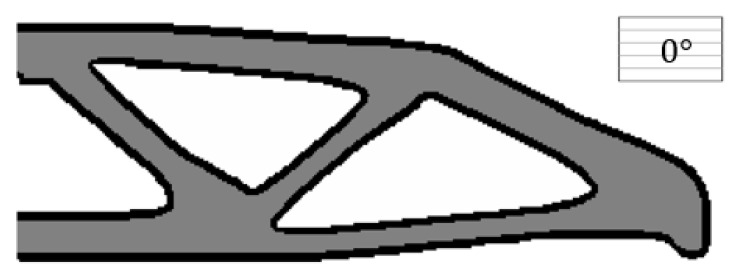
The optimized MBB beam.

**Figure 7 micromachines-11-00709-f007:**
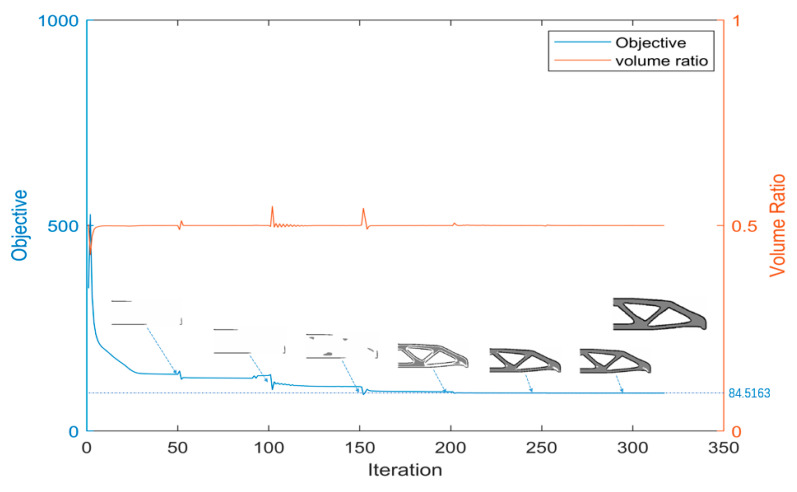
The convergence history.

**Figure 8 micromachines-11-00709-f008:**
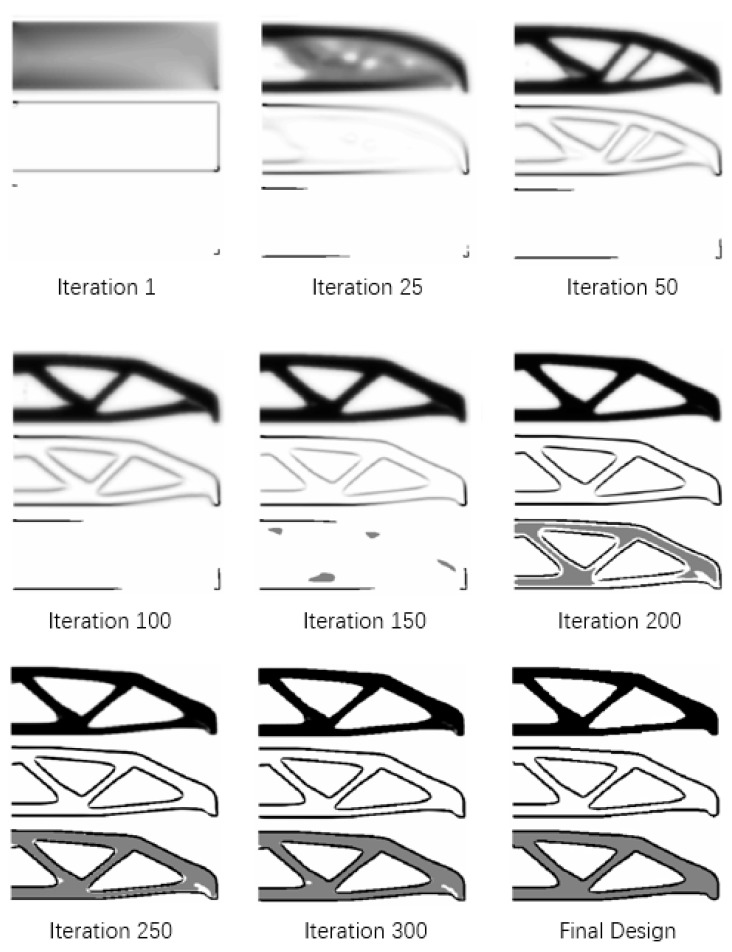
The topology of substrate domain ($\rho_{E}$) and boundary layer ($\bar{{\left\|\nabla \hat{\varphi_{E}}\right\|}_{\alpha }}$) evolution.

**Figure 9 micromachines-11-00709-f009:**
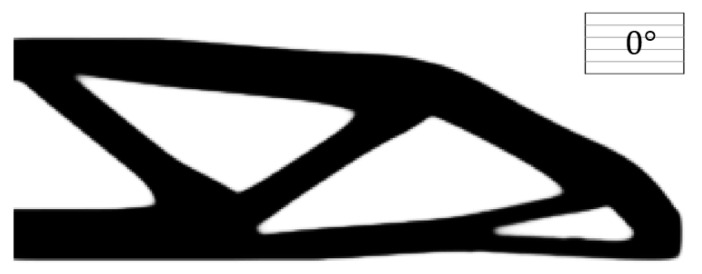
The optimized MBB beam without the boundary layer under the raster direction θ=0° (c=100.6204).

**Figure 10 micromachines-11-00709-f010:**
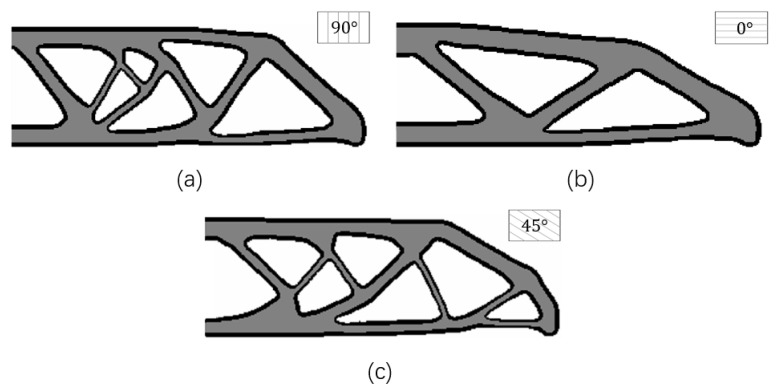
The optimized results with different raster directions: (**a**) θ=90°, c=130.4545; (**b**) θ=0°, c=88.3850; (**c**) θ=45°, c=132.0675.

**Figure 11 micromachines-11-00709-f011:**
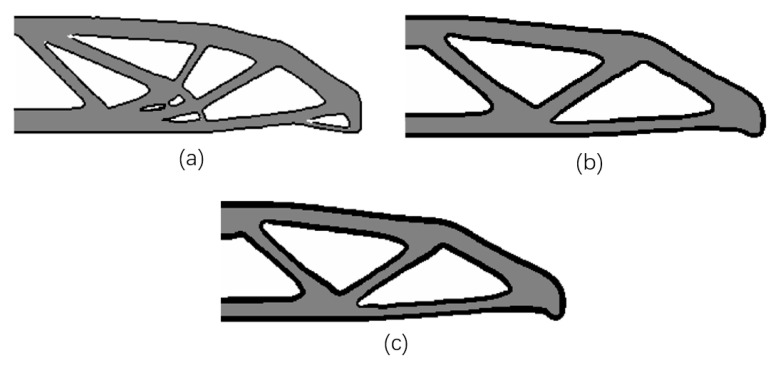
The optimized results with different boundary layer widths: (**a**) T=2, $R_{2}=5$, c=92.8804; (**b**) T=4, $R_{2}=10$, c=88.3850; (**c**) T=6, $R_{2}=15$, c=86.7072.

**Figure 12 micromachines-11-00709-f012:**
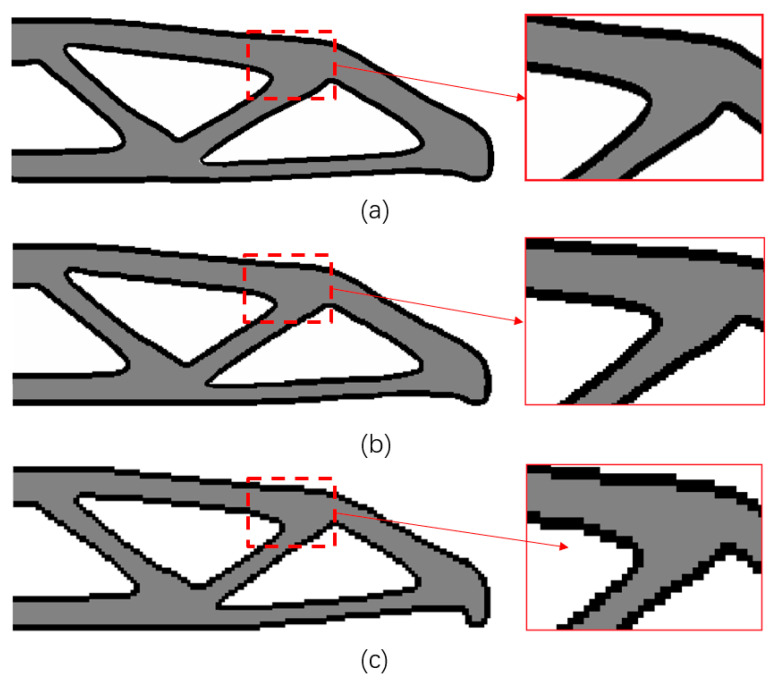
The optimized results with different mesh with the same boundary width: (**a**) 450×150 elements, C=83.0675; (**b**) 300×100 elements, C=88.3850; (**c**) 150×50 elements, C=97.2552.

**Figure 13 micromachines-11-00709-f013:**
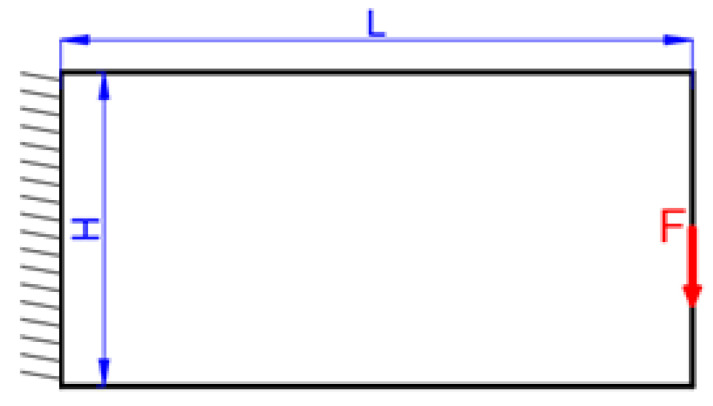
The cantilever beam.

**Figure 14 micromachines-11-00709-f014:**
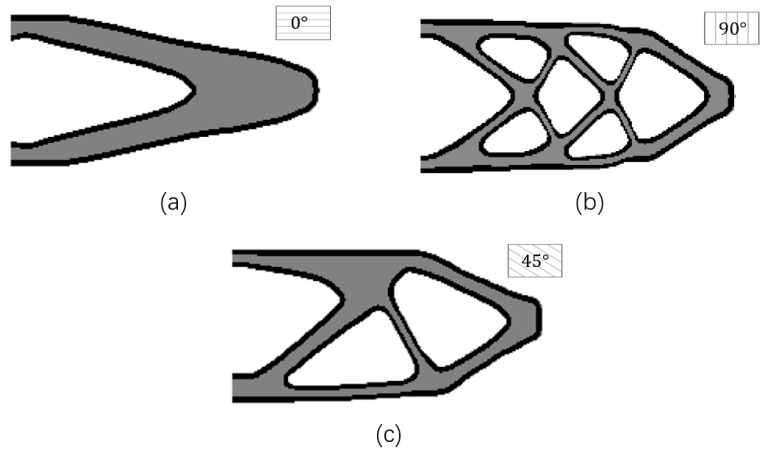
The optimized results with different raster directions: (**a**) θ=0°, c=36.9816; (**b**) θ=90°, c=52.7019; (**c**) θ=45°, c=42.0721.

**Figure 15 micromachines-11-00709-f015:**
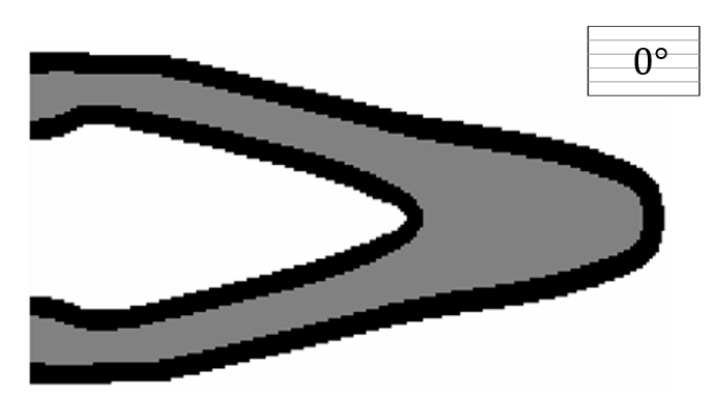
The optimized cantilever with a thick boundary layer T=10 under 200×100 elements: θ=0°, c=32.0089.

**Figure 16 micromachines-11-00709-f016:**
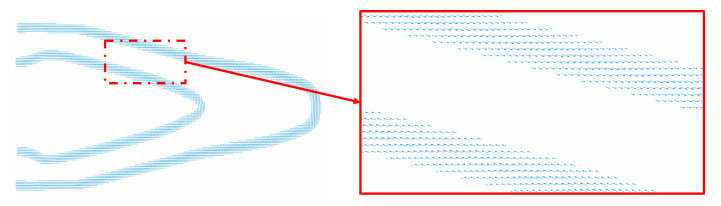
The boundary layer raster direction distribution.

**Figure 17 micromachines-11-00709-f017:**
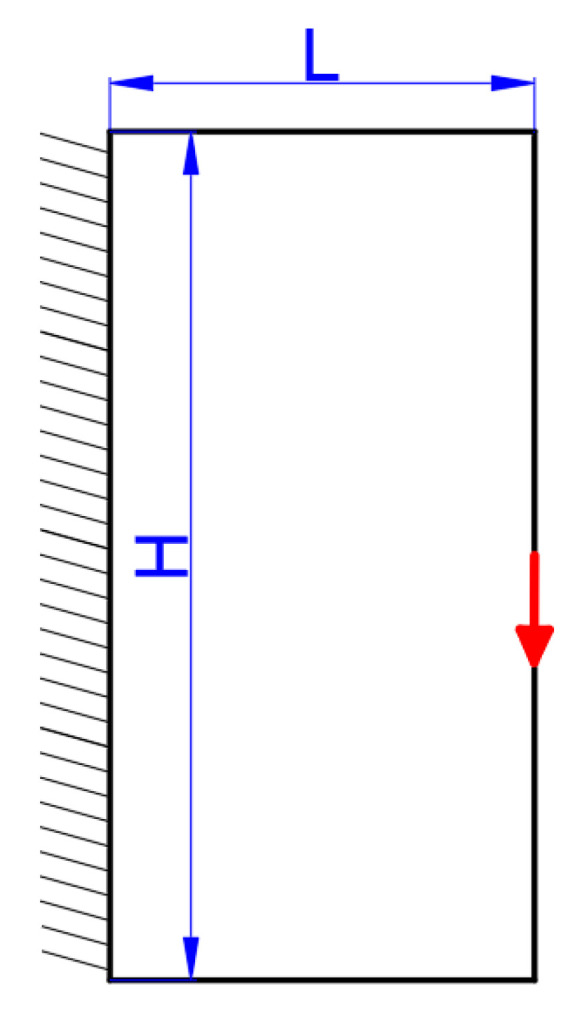
The short cantilever beam.

**Figure 18 micromachines-11-00709-f018:**
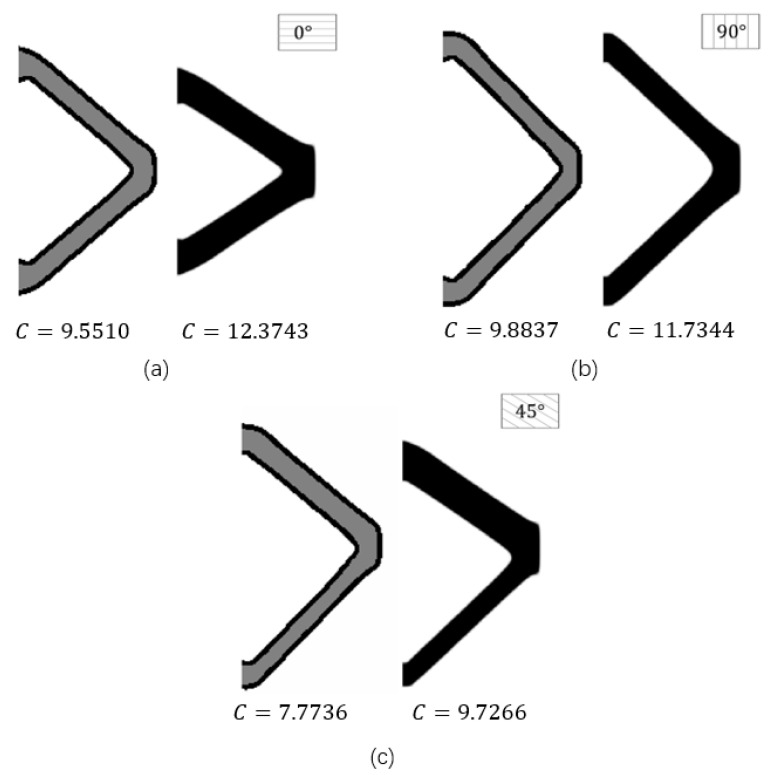
The optimized results with different raster directions: (**a**) θ=0°; (**b**) θ=90°; (**c**) θ=45°.

**Figure 19 micromachines-11-00709-f019:**
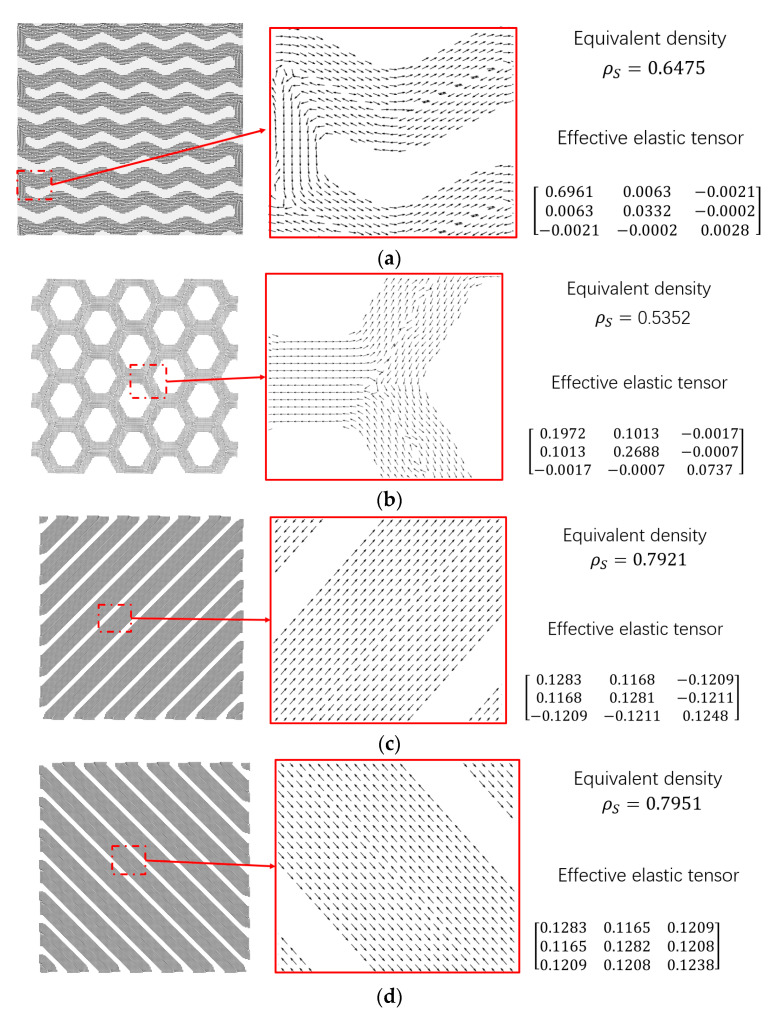
The different infill patterns. (**a**) wiggle, (**b**) honeycomb, (**c**) −45° line, (**d**) 45° line.

**Figure 20 micromachines-11-00709-f020:**
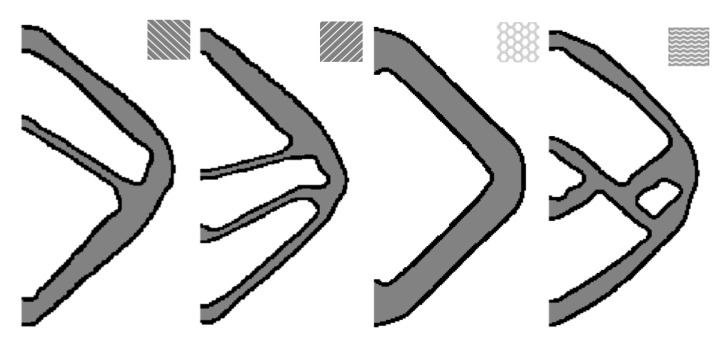
The optimized results with different infill patterns.

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
