# Peer review of "Topology Optimization for FDM Parts Considering the Hybrid Deposition Path Pattern"

_micromachines, 2020, doi:10.3390/mi11080709_

Round 1

Reviewer 1 Report

The authors develop an algorithm based on Solid Orthotropic Material with Penalization (SOMP) and DSP to find out an optimal structure design that considers HDP pattern and the anisotropic material properties. The external offset contour is assumed as  a  uniform  boundary  layer  structure  which  has  different  material  with  the  substrate  structure. The DSP method allows identifying the boundary layer and achieves the length control for it; and the anisotropic material topology optimization could be realized by the SOMP interposition model.

This is an interesting work with good scientific soundness. This Reviewer has some remarks to improve the manuscript, as follows:

  1. Although the authors took care to provide a good number of References, I suggest focusing on citing more recent papers on topology optimization issues and advances.
  2. In the work “Cross-section optimization of topologically-optimized variable-axial anisotropic composite structures”, the authors used an isotropic topology optimization approach for anisotropic fiber-reinforced structures and still obtained great improvements in terms of stiffness. How better their results could be if they would use your SOMP approach instead of SIMP? Please elaborate well on this and include this discussion in the Introduction section – this may increase the visibility of your work towards other communities, e.g. the composites community.
  3. Beginning of section 2.1: replace classic with classical
  4. In some Equations, the terms seem overlapped. See Eq 3, 4, 6, for instance.
  5. Optimization formulation: how do you define the volume reduction? Based on what assumptions? How do you determine the best relation between compliance and volume?
  6. Optimum topologies: please place side by side the optimized topology through SIMP and SOMP – then we can clearly see the potentialities and limitations of SOMP.
  7. Have you considered optimizing for minimum mass with stress constraints?

Author Response

Reply to Reviewer #1's comments

The manuscript (ID: micromachines-849058) has been carefully revised according to the reviewers’ comments. We appreciate their detailed and useful comments and suggestions. The point-by-point answers to the comments and suggestions are listed below.

Reviewer 1

For point 1 ‘Although the authors took care to provide a good number of References, I suggest focusing on citing more recent papers on topology optimization issues and advances.’:

Reply: Some new references have been already added in the revised manuscript.

For point 2 ‘In the work “Cross-section optimization of topologically-optimized variable-axial anisotropic composite structures”, the authors used an isotropic topology optimization approach for anisotropic fiber-reinforced structures and still obtained great improvements in terms of stiffness. How better their results could be if they would use your SOMP approach instead of SIMP? Please elaborate well on this and include this discussion in the Introduction section – this may increase the visibility of your work towards other communities, e.g. the composites community.’:

Reply: The relative content has been already added in the introduction part in revised manuscript. Several typical fiber-reinforced topology optimization works are discussed by us.

For point 3 ‘Beginning of section 2.1: replace classic with classical.’:

Reply: This grammar error has already been corrected (please see the revised manuscript).

For point 4 ‘In some Equations, the terms seem overlapped. See Eq 3, 4, 6, for instance.’:

Reply: These errors have already been corrected (please see the revised manuscript).

For point 5 ‘Optimization formulation: how do you define the volume reduction? Based on what assumptions? How do you determine the best relation between compliance and volume?’:

Reply: We assume the elemental mass is linearized with the elemental density; and the volume reduction is indirectly achieved by controlling the structural mass. The mass interposition is modified in the revised manuscript.

For point 6 ‘Optimum topologies: please place side by side the optimized topology through SIMP and SOMP – then we can clearly see the potentialities and limitations of SOMP.’:

Reply: The SOMP is not a novel method. Generally, the energy method combined with the filtering technology is used to determine the optimized fiber orientation and density simultaneously in each element.

For point 7 ‘Have you considered optimizing for minimum mass with stress constraints?’:

Reply: Stress constrained problem will be our future work content. Besides, our authors also have done some related work. Our initial idea is based on the current model, there will be two stress level interpositions: the projected density field and its normalized gradient field, and thus realizing the stress measure both in substrate domain and boundary domain. Because orthotropic material is considered in this model, instead of conventional von Mises criteria, Tsai Wu criteria is adopted. However, this method makes the whole model highly nonlinear, and therefore induce some numerical instability and hard to converge. This numerical issue is still needed to be solved by us.

Reviewer 2 Report

This paper presented a TO method, which introduced anisotropic materials to the coated structural design. The fiber path is assumed to be distributed along the outer shell while the fiber path in the substrate is fixed with a prescribed angle. While both the HDP approach and the implementation of anisotropic materials are not novel, the unique application is highly relevant to the AM process. The proposed design method can be of interests to industry practitioners. The numerical results demonstrated the effectiveness of the proposed method.

I recommend the acceptance of this paper if the authors can kindly address my concerns as follows.

1) Why not optimize the fixed angle for the substrate materials?

2) In section 4.1, please provide more details on the implementation of the asymmetrical boundary condition.

3) As for the implementation, did the authors use any commercial software to solve PDE state variables? If so, please list. If not, please clarify.

4) In figure 18, the direction arrows are too small to be seen.

5) I understand the authors may not be able to provide 3D examples. But please discuss potential challenges.

6) The fixed fiber path seems to be justifiable with the specific application discussed in this paper. But I noticed there are other recently published anisotropic TO works that discussed the optimization of build direction and even customized fiber path. Please conduct a search, and add related references.

7) In Eq.(6), the format is strange. Please fix.

8) In Eq.(42), the authors used \phi to denote the design field. But previously, it was \mu. Please be consistent.

9) The mass constraint discussed in Section 2.5 was not clear. Do you assume the substrate will be fully filled with unidirectional materials? In practice, the substrate will be gapped wall-like infills. In the latter case, the density of the substrate and shell will be different. Please clarify.

10) Following 9), it will be nice to introduce different settings of infill densities. This implementation gives much more flexibility to designers.

Author Response

Reply to Reviewer #2’s comments

The manuscript (ID: micromachines-849058) has been carefully revised according to the reviewers’ comments. We appreciate their detailed and useful comments and suggestions. The point-by-point answers to the comments and suggestions are listed below.

Reviewer 2

For point 1 ‘Why not optimize the fixed angle for the substrate materials?’:

Reply: In this work, we hope to emphasize the combination between the interface problem and SOMP method in topology optimization, and use these two schemes to realize HDP pattern-based optimization. So, the optimization of the substrate material is omitted by us. But in our future work, we will improve this aspect.

For point 2 ‘In section 4.1, please provide more details on the implementation of the asymmetrical boundary condition.’:

Reply: The corresponding details have already been provided in the revised manuscript.

For point 3 ‘As for the implementation, did the authors use any commercial software to solve PDE state variables? If so, please list. If not, please clarify.’:

Reply: The algorithm of this work is implemented in MATLAB, and all the PDE equations are solved through MATLAB.

For point 4 ‘In figure 18, the direction arrows are too small to be seen.’:

Reply: The figure has been already replaced with a higher DPI in the revised manuscript.

For point 5 ‘I understand the authors may not be able to provide 3D examples. But please discuss potential challenges.’:

Reply: Thanks for your understanding. In this work, we find that the desired interface width dictated by the mesh resolution, which means a coarse mesh resolution is impossible to identify a clear boundary layer. In 2-D case, we could simply use a finer mesh to achieve a better performance: the corresponding computational cost is acceptable, and our current algorithm could handle it well. However, in 3-D case, we found that a large-scale solver is needed: the efficiency of our algorithm is so slow, and the data memory is exceeded.

Because of the limitation of our research, we have not mastered the large-scale topology optimization technology yet. One feasible strategy for us is to integrate the commercial software (like ANSYS and so forth) with our algorithm, and we are trying to overcome this issue. The relative discussion has already added in the end of the revised manuscript.

For point 6 ‘The fixed fiber path seems to be justifiable with the specific application discussed in this paper. But I noticed there are other recently published anisotropic TO works that discussed the optimization of build direction and even customized fiber path. Please conduct a search, and add related references.’:

Reply: Some new references have been already added in the revised manuscript.

For point 7 ‘In Eq.(6), the format is strange. Please fix.’:

Reply: The format has already been corrected in the revised manuscript.

For point 8 ‘In Eq.(42), the authors used \phi to denote the design field. But previously, it was \mu. Please be consistent.’:

Reply: The format has already been corrected in the revised manuscript.

For point 9 and point 10 ‘The mass constraint discussed in Section 2.5 was not clear. Do you assume the substrate will be fully filled with unidirectional materials? In practice, the substrate will be gapped wall-like infills. In the latter case, the density of the substrate and shell will be different. Please clarify’; ‘it will be nice to introduce different settings of infill densities. This implementation gives much more flexibility to designers’:

Reply: Thanks for your reminding, In the initial manuscript, we assume the substrate material is fully filled with no gap, thus the densities of substrate material and boundary material could be seemed as the same. However, this assumption is not rigorous in practice. Therefore, we make some tiny modifications in the mass interposition.

Besides, in the revised manuscript, we add a new subsection (section 4.3.2) to provide several numerical cases under customized substrate deposition patterns with different infill densities. The effective mechanical properties of these infill pattern are predicted through energy-based homogenization method, while the rest part of algorithm remains unchanged.

Reviewer 3 Report

Excellent work, interesting topic, but lacks some practical aspect. Some detailed remarks:
1. Effectiveness of the presented method has been proven, but the efficiency was not - how long does it take to compute the optimal topology? What computing power is needed?
2. The paper relies on 2D cases (which I translates into considering a 3D print made of a single layer, or multiple 1D layer) and I believe that practicality of the approach can be only fully assessed when the 3D geometry case results are juxtaposed with results of experiments made on real 3D prints. Until that time, from a standpoint of 3D printing technology, this is just a preliminary result and an interesting insight. Experimental validation is a must (not mentioned by the authors).
3. What software was used for the computations?
4. Other minor remarks
- Fig. 13 - the drawing shows movable support, while the text says "clamped"
- Nearly 25% of references are 10 years old or older, that's merely acceptable. I'd add some more.

The paper is solidly written, although the presentation could use some more clarity here and there, especially in the first 2 sections. A clearly defined research purpose should be shown, as well as a review of basic assumptions, to make the paper more "reader-friendly" (took me 3 readings to get a clear picture, it shouldn't be like this). The English language is also solid, although some sentences could be polished (I recommend reviewing the whole paper for clarity of presentation and formulation of particular sentences.)

Author Response

Reply to Reviewer #3’s comments

The manuscript (ID: micromachines-849058) has been carefully revised according to the reviewers’ comments. We appreciate their detailed and useful comments and suggestions. The point-by-point answers to the comments and suggestions are listed below.

Reviewer 3

For point 1 ‘Effectiveness of the presented method has been proven, but the efficiency was not - how long does it take to compute the optimal topology? What computing power is needed?’:

Reply: The discussion of the algorithm effectiveness has already been added in the revised manuscript.

For point 2 ‘The paper relies on 2D cases (which I translates into considering a 3D print made of a single layer, or multiple 1D layer) and I believe that practicality of the approach can be only fully assessed when the 3D geometry case results are juxtaposed with results of experiments made on real 3D prints. Until that time, from a standpoint of 3D printing technology, this is just a preliminary result and an interesting insight. Experimental validation is a must (not mentioned by the authors).’:

Reply: In our future work, we will further provide 3D results. The relative conclusion and discussion have already been added in the end of the revised manuscript.

For point 3 ‘What software was used for the computations?’:

Reply: All the computations in this algorithm is achieved through MATLAB.

For point 4 ‘Other minor remarks: Fig. 13 the drawing shows movable support, while the text says "clamped"; nearly 25% of references are 10 years old or older, that's merely acceptable. I'd add some more.’:

Reply: These incorrected figures have already been changed and some new references are also added in the revised manuscript.

Round 2

Reviewer 1 Report

The paper can be now accepted for publication in Micromachines.